# Breakthrough COVID-19 Infections in the US: Implications for Prolonging the Pandemic

**DOI:** 10.3390/vaccines10050755

**Published:** 2022-05-11

**Authors:** Donald J. Alcendor, Patricia Matthews-Juarez, Duane Smoot, James E. K. Hildreth, Kimberly Lamar, Mohammad Tabatabai, Derek Wilus, Paul D. Juarez

**Affiliations:** 1Department of Microbiology, Immunology and Physiology, Center for AIDS Health Disparities Research, School of Medicine, Meharry Medical College, 1005 D.B. Todd Jr. Blvd., Nashville, TN 37208, USA; jhildreth@mmc.edu; 2Center for AIDS Health Disparities Research, Department of Microbiology, Immunology, and Physiology, School of Medicine, Meharry Medical College, 1005 D.B. Todd Jr. Blvd., Hubbard Hospital, 5th Floor, Rm. 5025, Nashville, TN 37208, USA; 3Department of Family & Community Medicine, Meharry Medical College, 1005 D.B. Todd Jr. Blvd., Nashville, TN 37208, USA; pmatthews-juarez@mmc.edu (P.M.-J.); pjuarez@mmc.edu (P.D.J.); 4Department of Internal Medicine, School of Medicine, Meharry Medical College, 1005 D.B. Todd Jr. Blvd., Nashville, TN 37208, USA; dsmoot@mmc.edu; 5Office of Health Disparities Elimination, Tennessee Department of Health, Nashville, TN 37243, USA; kimberly.lamar@tn.gov; 6School of Graduate Studies and Research, Meharry Medical College, 1005 D.B. Todd Jr. Blvd., Nashville, TN 37208, USA; mtabatabai@mmc.edu (M.T.); dwilus@mmc.edu (D.W.)

**Keywords:** SARS-CoV-2, COVID-19, coronavirus, breakthrough infections, vaccine, Delta variant, Omicron variant

## Abstract

The incidence of COVID-19 breakthrough infections—an infection that occurs after you have been vaccinated—has increased in frequency since the Delta and now Omicron variants of the SARS-CoV-2 coronavirus have become the dominant strains transmitted in the United States (US). Evidence suggests that individuals with breakthrough infections, though rare and expected, may readily transmit COVID-19 to unvaccinated populations, posing a continuing threat to the unvaccinated. Here, we examine factors contributing to breakthrough infections including a poor immune response to the vaccines due to the fact of advanced age and underlying comorbidities, the natural waning of immune protection from the vaccines over time, and viral variants that escape existing immune protection from the vaccines. The rise in breakthrough infections in the US and how they contribute to new infections, specifically among the unvaccinated and individuals with compromised immune systems, will create the need for additional booster vaccinations or development of modified vaccines that directly target current variants circulating among the general population. The need to expedite vaccination among the more than 49.8 million unvaccinated eligible people in the US is critical.

## 1. Introduction

Severe acute respiratory syndrome coronavirus 2 (SARS-CoV-2) is a newly emerged coronavirus that has reached pandemic levels since March 2020 [1,2,3,4]. SARS-CoV-2, the virus that causes COVID-19, may produce asymptomatic, as well as severe, acute disease with life-threatening consequences, particularly in medically underserved and vulnerable individuals with underlying comorbidities [5,6]. As of 13 March 2022, the confirmed number of COVID-19 cases in the United States (US) was 79.5 million (NY Times https://www.nytimes.com/interactive/2021/us/covid-cases.html accessed on 13 March 2022). The number of deaths attributed to COVID-19 was 968,320. In addition, 557 million vaccinations have been administered, 254.4 million people have received at least one dose of a COVID-19 vaccine, and 217 million people are fully vaccinated (65.8% of the population) [7] (NY Times https://www.nytimes.com/interactive/2021/us/covid-cases.html accessed on 13 March 2022); however, the level of vaccine hesitancy and resistance has remained high throughout the US since the onset of the pandemic, especially in the South [8]. Vaccine hesitancy and resistance has been particularly high among African Americans, Latinx, and rural residents due to the fact of a history and legacy of racial injustices, social inequities, and negative experiences within a long-standing culturally insensitive health care system [9]. The level of vaccine hesitancy and resistance in the US is high in rural counties, in the South and Midwest. In these counties, COVID-19 vaccine resistance is entrenched most among individuals who identify as White, rural, Republican, and evangelical Christian [8].

Breakthrough COVID-19 infections, or post-vaccination infections, occur when an individual has been vaccinated completely and over time acquires a new infection from the COVID-19 virus/most recent circulating variant [10]. These infections are referred to as “breakthrough infections”, or post-immune infections, due to the ability of the COVID-19 virus to break through the barrier of immune protection provided by the vaccines [11,12]. Breakthrough infections were expected, as COVID-19 vaccines were never shown to be 100% effective against infection, and immune protection after vaccination may vary among individuals [13] and wanes over time. The concern with the emergence of new variants, such as Omicron, now dominant in the US, is the ability of these mutants to evade vaccine-induced immunity and cause asymptomatic—and, although rare—severe and life-threatening disease [14]. In this discussion, we examine the dynamics of breakthrough infections amid the emerging, dominant Omicron variant and its displacement of the Delta variant in the US. Breakthrough infections are known to exist due to the Delta variant’s contribution to COVID-19 disease and death [15,16]. In this study, we provide an overview COVID-19 virus replication and its contribution to breakthrough infections in the Betacoronavirus family, with recent studies on breakthrough infections in both communal and health care settings. We also examined the genetic characteristics of the viral variants that avoid the immune response and contribute to breakthrough infections. Further, we identified important reasons and underlying conditions that may contribute to the current rise in these infections. Finally, strategies are provided that could be implemented to reduce the number of breakthrough infections.

## 2. Betacoronaviruses

Coronaviruses (CoVs), positive-sense, single-stranded, enveloped, RNA viruses that belong to the subfamily Coronavirinae, family Coronavirdiae, and order Nidovirales, are classified into four genera of CoVs: Alphacoronavirus (αCoV); Betacoronavirus (βCoV); Deltacoronavirus (δCoV); Gammacoronavirus (γCoV) [17,18]. To date, five βCoVs (HCoV-OC43, HCoV-HKU1, severe acute respiratory syndrome CoV (SARS-CoV), Middle East respiratory syndrome CoV (MERS-CoV), and most recently βCoV SARS-CoV-2 (COVID-19) have been discovered [19,20,21,22]. Human coronaviruses, HCoV-229E, and HCoV-OC43, have long been known to circulate among global populations as early as the 1960s. Together with the more recently identified HCoV-NL63 and HCoV-HKU1, the longstanding CoVs are usually associated with mild respiratory tract infections related to the common cold. The βCoV lineage, HCoV-OC43 and HCoV-HKU1, is generally associated with self-limiting, upper respiratory infections in immunocompetent hosts and occasionally lower respiratory tract infections in immunocompromised hosts and the elderly [23]. The World Health Organization (WHO) has classified COVID-19 as a βCoV of group 2B [24]. CoVs cause respiratory, enteric, hepatic, and neurological diseases in different animal species including camels, cattle, cats, and bats [23]. Coronaviruses possess the largest genomes of all RNA viruses, consisting of approximately 30 kb. COVID-19 belongs to the βCoVs genera and has 89% nucleotide identity with bat SARS-like-CoVZXC21 and 82% with that of human SARS-CoV [24]. Examination of the viral evolution shows that bats and rodents are gene sources for most αCovs and βCoVs, while avian species are the proposed gene sources of most δCoVs and γCoVs [25]. CoVs have been found to cross species barriers to infect humans and have emerged to cause significant morbidity and mortality in the general population. The most recent examples are the SARS-CoV, which emerged in China in 2002 with 8000 infections and 800 deaths [26,27], and the MERS-CoV, which emerged in the Arabian Peninsula in 2012 [28,29,30]. For SARS-CoV-2, the zoonotic intermediate host remains unknown, and the evolution of viral variants will continue, particularly among the unvaccinated and vaccinated populations with breakthrough infections. Zoonotic coronavirus infections that become adapted to humans via zoonotic intermediate hosts [31] could produce novel pathogens, leading to more pandemics.

## 3. Coronavirus Replication

Briefly, to initiate infection, the SARS-CoV-2 viral particle binds to the cognate receptor, angiotensin-converting enzyme 2 (ACE2), together with the host cell surface serine protease TMPSSR2. (Figure 1) [32]. This coordinated binding induces membrane fusion and promotes viral uptake and fusion at the endosomal membrane (Figure 1). Viral uncoating occurs and the genomic RNA is released into the cytoplasm, allowing immediate translation by host cell polyribosomes of two major open-reading frames (ORFs): ORF1a (encoding pp1a) and ORF1b (encoding pp1b) (Figure 1). The pp1a and pp1b polyproteins are co-translated and post-translationally processed into individual non-structural proteins forming viral replication and transcriptional complex (Figure 1). Replication and transcription sub-genomic mRNAs are represented as nested sets of coronavirus mRNAs encoding both structural and accessary proteins [32]. Translated structural proteins translocate to the ER, where they interact with N-encapsidated + sense genomic RNA (Figure 1). After encapsidation, virion egress is accomplished via exocytosis (Figure 1). It should be noted that the SARS-CoV-2 RNA genome involves the core replication–transcription complex (RTC, nsp12–nsp7–nsp8) and the proofreading complex (nsp14–nsp10) that can correct mismatched base pairs during replication. However, mutations do occur but not often as in the case of other RNA viruses without repair mechanisms.

## 4. Viral Variants

SARS-CoV-2 mutants that develop with the fitness to replicate and transmit sufficiently have the potential to be novel variants of concern (VOCs). VOCs are associated with having enhanced transmissibility or virulence, reduction in neutralization by antibodies obtained through natural infection or vaccination, the ability to evade detection, and a decrease in therapeutic or vaccination effectiveness [33]. As of 11 December 2021, five VOCs have been identified since the beginning of the pandemic using recent epidemiological data from the World Health Organization (WHO) [34]. The VOCs include Alpha (B.1.1.7), first reported in the United Kingdom (UK) in late December 2020; Beta (B.1.351), first reported in South Africa in December 2020; Gamma (P.1), first reported in Brazil in early January 2021; Delta (B.1.617.2), first reported in India in December 2020; Omicron (B.1.1.529), first reported in South Africa in November 2021 [34]. These variants have multiple mutations, with many occurring at the receptor-binding domain (RBD) locus of the S1 subunit of the virus spike glycoprotein, which is the cognate receptor that binds to the ACE2 host protein to initiate viral infection (Figure 2) [35]. The virus accumulates mutations in the RBD to achieve weakened antibody recognition while retaining ACE2 binding, and by selection, only some of these variants will achieve fitness to replicate and be transmitted. The number of variants will likely increase overtime as long as viral transmission continues to occur in the general population. Some of these mutations, such as the D164G mutation in the spike protein domain, may enhance viral fitness [36]. However, other mutations exist that will decrease viral fitness, reduced virulence, and are not transmitted. Functional genetic variants emerge via selection of those mutations resulting in viable viral progeny that may be transmitted to a susceptible host. Those mutations resulting in viral progeny that are dysfunctional lose the capacity to replicate and will not be transmitted. Mutants that are transmitted frequently have an entry or a replication advantage. They often replicate to higher titers than the parent strain and may cause more severe disease and death when compared with the parental strain. COVID-19 variants continue to evolve, and through this selective process of fit mutations that are replication-competent, it is highly likely more mutant strains will arise with the potential for hybrid variants via dual infection, leading to a novel strain of high virulence that is unaffected by existing vaccines. The Delta and Omicron variants are highly transmissible, and with the absence of preexisting immunity in the global population, their natural disappearance appears unlikely. This will require novel vaccine strategies for development of mRNA vaccine cocktails as part of a vaccine formulation.

### 4.1. The Delta Variant

The Delta variant was first identified in India in December 2020. The highly contagious variant had become the predominant mutant strain of the SARS-Cov-2 virus, at the time accounting for more than 90% of new COVID-19 cases in the US, until the appearance of the Omicron variant in December 2021 [37]. However, the Omicron variant rapidly overtook the Delta variant, (increasing from 1% to >50% of circulating viral lineages) during a two-week period. The Delta variant was associated with severe disease and death and is included here [38]. During the Delta wave, masking and travel restrictions in countries around the world were reinstated, reinforcing the need to vaccinate the global population. COVID-19 vaccines are largely protective against this variant preventing severe disease, hospitalizations, and death; however, due to the volume of new infections, the number of breakthrough infections due to the transmission of the Delta variant was highly significant [39,40]. This particular variant harbors several mutations that uniquely have given it the ability to spread more easily than previous COVID-19 variants [41]. The variant appears to have a higher replication rate when compared with previous variants at the time, and a recent study suggests that the Delta variant produces higher systemic viral loads that are 1000 times higher than the original strain of virus identified in Wuhan, China [42]. Testing data showed that cycle threshold value, which is the number of amplification cycles needed for the virus to be detected, is lower in specimens from individuals infected with the Delta variant, which means their systemic viral load is higher [43]. Higher viral loads may allow the virus, suited for infecting airway cells, to induce human transmissions after lower durations of exposure.

The Delta variant has a unique assortment of mutations in the spike glycoprotein known to bind to the ACE-2 entry receptor on some human cells [44]. These mutations lead to structural changes that alter the sidechain conformation to weaken the interactions with antibodies. The mutations may also lead to diagnostic detection failures [45]. According to the US Centers for Disease Control and Prevention (CDC), signature spike mutations in the aggregated Delta and Delta Plus variants include T19R, (V70F*), T95I, G142D, E156-, F157-, R158G, (A222V*), (W258L*), (K417N*), L452R, T478K, D614G, P681R, and D950N (* exclusively expressed in the Delta Plus variant) [46,47]. The difference between the Delta and Delta Plus variants is based on the K417N mutation in the parent Delta variant [48]. The Delta variant has an L452R mutation that is predicted to allow the virus to evade antibody neutralization, and interestingly, it lacks the E484K mutation found among other variants that are thought to accomplish the same function in a less efficient manner. Although the Delta variant, otherwise referred to as the B.1.617.2 variant of SARS-CoV-2, was identified in India in late 2020. The Delta variant was named on 31 May 2021 and spread to over 179 countries by 22 November 2021 before being displaced by the Omicron variant in December 2021 [49,50,51]. The Delta variant was responsible for 100% of new infections in the US until the emergence of the Omicron variant. During the predominance of the Delta variant, a surge in new infections occurred, and its emergence coincided with a sharp rise in hospitalizations and an increase in pediatric hospitalizations [52]. The Omicron variant was found to have a higher rate of transmission than other variants and has been found to be highly transmissible in public indoor settings and households [53]. Individuals infected with the Delta variant have been predicted to transmit the virus at levels equal to individuals infected who were unvaccinated [54]. Moreover, the Delta variant appears to be approximately 60% more transmissible than the previously highly infectious Alpha variant (B.1.1.7), identified in the UK in late 2020 [55]. Due to the fact of vaccine inequities around the world, the Delta variant posed the greatest risk to low-income countries with limited access to vaccines, particularly those countries in Africa, where most nations have fully vaccinated less than 13% of their populations [56]. (African CDC, https://africacdc.org/covid-19-vaccination/ accessed on 12 March 2022) Therefore, the Delta variant poses an ongoing risk to global public health.

### 4.2. The Omicron Variant

The current Omicron variant has more than 30 mutations in the spike protein, with 15 of these mutations in the RBD [57]. Studies by Shau et al. show that mutations in the RBD of the Omicron variant bind ACE2 at a rate 2.5 times stronger than prototype SARS-CoV-2. The binding affinity has been localized to three substitutions (i.e., T478K, Q493K, and Q498R), which significantly contribute to the binding energies and electrostatic potential of the RBD Omicron–ACE2 complex [57]. The Omicron variant also harbors the E484A substitution that allows the virus to escape from neutralizing antibodies [57]. The E484A mutation in the RBD has been associated with immune escape from approved COVID-19 therapeutic antibodies [57]. In a study by Bushman et al., the Omicron variant (B.1.1.529) showed enhanced transmissibility and partial immune escape [58]. Omicron variant-based three-dimensional structures of antibody–RBD complexes may be greater than ten times more contagious than the original virus or about twice as infectious as the Delta variant [59]. In another study, investigators employed a mathematical model to simulate the dynamics of wild-type and variant strains of SARS-CoV-2 in the context of vaccine rollout and nonpharmaceutical interventions [60]. They concluded that when variants with acquired phenotypes of enhanced transmissibility and partial immune escape (Omicron/Delta) combine, the variant has the potential to continue spreading, even as immunity in the population increases, thereby limiting the impact of vaccinations [60]. This is, however, a theoretical model that will require validation. It has been observed that full vaccination is associated with reduced risk of COVID-19 breakthrough infections [60]. The Omicron variant has been observed to be more likely to escape current vaccines than the Delta variant as well as to escape protection from existing US Food and Drug Administration (FDA)-approved monoclonal antibodies [61]. With regard to the Omicron variant, the term “breakthrough infection” has recently been challenged by the increased infection rates among healthy, highly vaccinated populations. This is unexpected because the underlying reasons for acquiring breakthrough infections is waning immunity, age, and underlying comorbidities which has been circumvented by the Omicron variant when compared to previous variants, although often resulting in mild disease. Variant lineages that evolve to escape immune protection from the original spike protein antigen, which all currently approved vaccines in the US are based upon, should be carefully monitored. These findings support the future development of virus specific and viral combinatorial vaccines. 

## 5. Breakthrough Infections (Post-Immune Infections)

It should be known that the CDC tracks breakthrough infections in the US only. The agency only refers to those individuals who have tested positive for COVID, have been hospitalized, have developed severe illness from the disease, or have died after being vaccinated fully [62]. Therefore, the true number of post-vaccination infections (breakthrough infections) are unknown and highly underrepresented in the US. In addition, as the number of people who get vaccinated increases, the number of post-vaccination infections also will increase. The CDC states that a vaccine breakthrough infection is defined as the detection of SARS-CoV-2 RNA or antigen in a respiratory specimen collected from a person ≥ 14 days (about 2 weeks) after they have completed all recommended doses of the FDA-authorized COVID-19 vaccine [63]. The dynamics of breakthrough infections involve several factors, including an individual’s response to the vaccine regime, the virulence and viral load of the variant they are expose to, the duration of a protective level of neutralizing antibodies, and T-cell immune response (Figure 3). In addition, preexisting conditions varying from one individual to the next, contributing to breakthrough infections including but not limited to immune suppression, age, genetics, and a variety of other underlying comorbidities (Figure 3). The role of the Omicron variant in the development of breakthrough infections is unclear, owing to the mild disease observed in some patients (Figure 3).

As breakthrough infections may be asymptomatic and must be documented by positive test results, illness, and/or hospitalizations, they are highly underreported. These breakthrough infections have fostered anxiety and confusion for vaccinated populations within the US who were convinced that complete vaccination would protect them from infection. These breakthrough infections also have inadvertently added to the increasing level of vaccine hesitancy and resistance regarding the safety and efficacy of the vaccines in the US. Despite high-level efficacy provided by the current vaccines (Pfizer-BioNTech, Moderna, and Johnson & Johnson) and the recommended boosters [64,65,66], breakthrough infections have occurred and will continue. Breakthrough COVID-19 infections are rare in occurrence, but due to the reported increase in viral loads associated with breakthrough infections and the greater risk of transmission to unvaccinated populations, further studies are needed. The rise in breakthrough infections associated with the current dominant Omicron strain, the 49.8 million people in the US that remain unvaccinated, the increased new infections, the recent surge in hospitalization due to the Omicron variant, and deaths-along with stagnant vaccination rates, specifically in the South and Midwest due to the fact of vaccine hesitancy and resistance, taken together represents an unmanageable public health crisis. We are aware that the rates of new infection at the time of this writing have decreased; still, we must proceed with caution and not terminate CDC COVID-19 mitigation guidelines. The CDC, however, has reported that COVID-19 breakthrough infections typically do not result in severe disease and are uncommon among individuals; further, the agency reports that current vaccines are effective in preventing severe disease, which could lead to hospitalizations and deaths. Overall, it is still unclear how post-vaccination infection will impact the COVID-19 pandemic in the US, as some post-vaccination infections are asymptomatic, with the potential to transmit infections to people who are unvaccinated or have immunosuppressive medical conditions.

In April 2021, the CDC reported approximately 5800 breakthrough infections for 77 million vaccinated persons. This was the CDC’s first public accounting of breakthrough cases [67]. The agency reported COVID-19 vaccine breakthrough infections among all people of all ages eligible for vaccination at that time. However, slightly more than 40% of the infections were in people aged 60 or older. It was determined that 65% of the breakthrough cases were female, and 29% of the cases were asymptomatic [67]. In a recent study, Juthani et al. used data collected by the Yale New Haven Health System to examine the incidence of hospitalization among COVID-19 breakthrough infections that were confirmed by a positive PCR test for SARS-CoV-2 at the time of admission; the incidence of severe or critical COVID-19 illness remained low [68]. Researchers observed that among 54 patients who were vaccinated fully with Pfizer-BioNTech/BNT162b2 or Moderna/mRNA-1273 or a single dose of Johnson & Johnson-Janssen/Ad.26.COV2.S and who experienced breakthrough infections and were hospitalized: 25 (46%) patients were asymptomatic (admitted to hospitals for non-COVID-19-related diagnoses but with an incidental positive PCR test for SARS-CoV-2); 4 (7%) had mild disease; 11 (20%) had moderate disease; 14 (26%) had severe or critical illness [68]. However, among those who were critically ill, the average age was 80.5 years. Several comorbidities existed among these 14 patients, including overweight, cardiovascular disease, lung disease, malignancy, and type 2 diabetes as well as the use of an immunosuppressive agent. Furthermore, 13 out of 14 critically ill patients received the Pfizer vaccine [68]. Interestingly, in a study performed in Israel by Bergwerk et al., which examined COVID-19 vaccine breakthrough infections among 1497 fully vaccinated health care workers, researchers determined that 39 workers experienced breakthrough infections [69] (Figure 4). The researchers matched uninfected controls to patients with breakthrough infections; these patients had antibody titers obtained within one week of a positive test (peri-infection period) [69] (Figure 5). Bergwerk et al. observed neutralizing antibody titers in infected patients during the peri-infection period were lower than those in matched uninfected controls [69]. They concluded a correlation existed with peri-infection neutralizing antibody titers and the incidence of breakthrough infections in their cohort [69] (Figure 4).

## 6. Potential Causes and Circumstances Surrounding Breakthrough Infections

Following the Emergency Use Authorization (EUA), FDA-approved COVID-19 vaccines are observed to be highly effective with efficacies of 95% (Pfizer-BioNTECH), 94% (Moderna), and 74% (Johnson & Johnson). Some side effects and FDA warnings are related to these vaccines, but the benefits far outweigh the risks [70]. Primarily, COVID-19 vaccines were never advertised to be 100% effective, and the vaccines are not expected to produce a protective immune response in all recipients. The natural waning of the immune response to specific antigens in a vaccine may vary from one individual to the next. The dominant and highly transmissible Delta and Omicron variants that induce higher viral loads contribute to the rise in breakthrough COVID-19 infections. The duration of immune protection afforded by a vaccine also is unknown and is dependent on many factors, including age, gender, underlying clinical conditions, genetics, and the employed vaccine.

The Delta variant has a specific mutation, P681R (Pro681 Arg), located at the furin cleavage site [71]. This mutation has been associated with increased cleavage of spike into S1 and S2 fragments [72]. The functional role of the P681R mutation could be related to other spike mutations at the amino-terminal end of the S1 subunit, where a deletion in the Delta variant of amino acids 157–158 occurs as well as mutations at T19R, G142D, and R156G. These mutations could lead to substantial rearrangement of the N-terminal domain that may have allosteric effects on the RBD, which induces binding with additional cellular receptors, thereby leading to increased infectivity.

Mutations in the spike protein that increase infectivity could enable viral attachment for more efficient infection of respiratory epithelial cells, avoiding the poor neutralizing antibodies in the mucosa (depicted in Figure 5). More importantly, the Delta variant spike protein is most efficient at establishing membrane fusion when compared with other variants [73] (depicted in Figure 5). This ability of the Delta variant spike protein to efficiently induce the development of multinucleated cells or syncytia could enable the virus to disseminate from one cell to another without needing to exit the cell, avoiding exposure to neutralizing antibodies (depicted in Figure 5). This could contribute to the increased number of breakthrough infections seen after infection by the Delta variant.

### 6.1. Waning of Immune Protection

Waning of immune protection is a natural phenomenon that varies from one individual to the next. More often, when waning of immune protection is discussed, most of the discussion focuses on antibody production. However, immune protection against virus infection induced by vaccinations is mediated through a complex interplay between innate, humoral, and cell-mediated immunity. The waning of the immune response does not involve all components required for immune response protection. T-cell and B-memory cell functions may not be affected when compared with waning of neutralizing antibodies to COVID-19. The antibody protection level to prevent COVID-19 clinical disease is unknown; therefore, the waning of the immune response to the disease remains undefined. For the most vulnerable populations, waning of the immune response would suggest an increased risk for severe COVID-19 disease, and current booster vaccinations have become warranted for individuals who are immunocompromised.

### 6.2. Aging and Immune Responses to Vaccines

Aging is a risk factor for a poor response to antigens in vaccines [74,75,76]. Age-related decline and dysregulation of immune function, such as immunosenescence and inflammaging, could increase an individual’s vulnerability to COVID-19, as well as provide inefficient protection after vaccination [77]. The quality and quantity of antibody responses, including specificity and class of antibody produced, is affected by aging [78]. Defects in T cells are a main reason for age-related immune dysfunction [78]. Impairments in stem cells and lymphoid progenitors in the bone marrow and thymus, reductions in clonal diversity of naive T cells, and the skewing of their function to a Th1 pro-inflammatory role may result in the reduction of immune responsiveness to vaccines in the elderly [78]. In addition, follicular CD4 T-cell functions that are suboptimal for proper B-cell maturation play a role in changes that may occur regarding antibody production [79]. Decline in B-cell-specific transcription factors, such as activation-induced cytidine deaminase expression along with suboptimal T-cell helper functions and B-memory cell switching, results from aging [80]. Moreover, aging may result in alteration of innate immune response pathways, including pathogen-associated molecular patterns, to engage pattern recognition receptors on immune effector cells (macrophages, monocytes, and neutrophils) that lead to the induction of type I interferon (IFN) responses, which mediate anti-viral immunity. Alterations in innate immune recognition pathways also may contribute to inflammaging, which is the chronic production of proinflammatory cytokines (IL-6, TNFα, and type I IFN) [81]. Taken together, this may lead to decreased vaccine and viral infection responses in the elderly. Strategies targeting dysfunctional innate and adaptive immune pathways with innovative adjuvant therapies could produce enhanced immune responses among the elderly receiving COVID-19 vaccines, hence, reducing the number of breakthrough infections within this population. In the short term, COVID-19 booster vaccinations may be required for elderly individuals. The adaptive immune system is critical to the clearance of all viral infections and the interactions of the adaptive immune cell components made up of B cells, CD4+ T cells, and CD8+ T cells are essential for immune protection. The SARS-CoV-2 coronavirus represents a formidable challenge for the innate immune system. By inhibiting type I and type III interferon (INFs) responses, the virus can delay activation of the adaptive immune responses [82]. Impaired or delayed type I and type III INF responses have been associated with more severe complications of COVID-19 disease [83]. Even more SARS-CoV-2 specific T-cell responses have been associated with mild disease [84]. Therefore, individuals with defective IFN response may be predisposed to breakthrough infection. This delay in priming of the adaptive immune response can allow virus spread in the upper airway and lungs that could lead rapid changes in pathology requiring hospitalization and intensive care interventions. Suppression of the Type 1 IFN receptor gene IF NAR2 has been associated with severe COVID-19 disease in a genome-wide association study, which correlates with other studies that observe the significance of IFNs in controlling infections [85]. Both Th1 cells and T-follicular helper cells are produced via CD4+ T-cell differentiation after virus infection. Th1 cells have antiviral properties via the production of cytokines and γ-interferon and T-follicular helper cells provide B cell help and are important for the development of most neutralizing antibody responses, as well as memory B cell responses [86]. Studies have shown no correlation in neutralizing antibodies and COVID-19 disease severity [87] however reduced disease severity with the frequencies of circulating T-follicular helper cells produced during acute phase COVID-19 disease has been observed [88]. This would suggest that the level or frequency of circulating T-follicular helper cells post-vaccination could be a risk factor for breakthrough infections. The presence of CD8+ T cells is also associated with better clinical outcomes due to the fact of COVID-19 disease [89]. The effects of CD8+ T cells are observed early during acute phase disease and has been associated with production of factors with antiviral properties including cytokines, IFN gamma, granzyme B, perforin, and CD107a [90]. The absence of CD8+ T cells early in the acute phase of the disease could increase the risk of breakthrough infections. 

### 6.3. Individuals with Weakened Immune Systems and COVID-19 Vaccines

Differences in race, gender, age, and underlying medical conditions may produce different profiles of antibodies from the same vaccine. Individuals with genetic, clinical conditions, or medical treatments that weaken their immune systems are more likely to have breakthrough infections after COVID-19 vaccinations. Genetic conditions and autoimmune diseases may alter your immune response to the vaccines. However, the American College of Rheumatology COVID-19 Vaccine Clinical Guidance recommends that people with autoimmune and inflammatory rheumatic diseases-which includes lupus-get the vaccine unless they have an allergy to one of the ingredients [91]. Immunosuppression is required for allograft transplantation, and solid organ transplant recipients are at higher risk for the more severe symptoms of COVID-19. In this vulnerable population, immunosuppression may severely impact the immune response to current and future vaccines. However, recommended guidelines reported as Guidance for Cardiothoracic Transplant and Mechanical Circulatory Support Centers regarding SARS-CoV-2 infection and COVID-19 have been established and must be followed when performing heart transplantation [92,93]. A recent study conducted by Aslam et al. found that solid organ transplant recipients who received a vaccine for COVID-19 showed significant protection against the disease [94]. During the study, researchers observed an 80% reduction in the incidence of symptomatic COVID-19 when compared with unvaccinated solid organ transplant recipients. In addition, no COVID-19-related deaths occurred in the four breakthrough infections among patients who received the COVID-19 vaccine [94]. Currently the FDA/CDC guidance includes recommendations that for people ages 12 years and older who are moderately or severely immunocompromised should receive a total of four doses of COVID-19 vaccine as a precaution to boost their immunity in response to the highly transmissible Omicron variant, currently the dominant strain in the US.

### 6.4. Individuals Receiving Biologic Immunosuppressive Therapies

Therapeutic biologics that target immune cell functions and inflammatory cytokines are now in routine use in the clinic for conditions such as lupus, multiple sclerosis, rheumatoid arthritis, and psoriasis. Individuals who receive biologic immunosuppressive therapies originally were at higher risk for severe symptoms of COVID-19 disease; however, biologic immunosuppressant therapy was not observed to increase risk of SARS-CoV-2 infection or COVID-19 severity [95]. A retrospective study conducted using institutional data and the Massachusetts state public health registries examined 7361 patients prescribed biologic therapy from July 2019 through February 2020, matching them to 74,910 controls [96]. The data were then cross-referenced with COVID-19 infection and all-cause mortality data from the Massachusetts Department of Public Health from 1 March 2020 to 19 June 2020 [87]. The study found 87 (1.2%) COVID-19 infections and seven (8.0%) deaths in patients treated with biologics, compared with 1063 (1.4%) infections and 71 (6.7%) deaths in the control group [96]. Finally, the study revealed no evidence that biologic immunosuppressants increase the risk of COVID-19 disease severity and mortality [87].

### 6.5. Genetics and Breakthrough Infections

Omics technologies have not been fully explored to uncover genetic predictors of COVID-19 disease; however, in a study by Carapito et al. that examined multi-omics analysis combined with artificial intelligence in a young patient cohort that included both critically ill and non-critically ill patients with COVID-19 [97], whole-genome sequencing, whole-blood RNA sequencing, plasma and blood mononuclear cell proteomics, cytokine profiling, and high-throughput immunophenotyping were included in the analysis [97]. The study revealed upregulation of the metalloprotease ADAM9 in critical ill patients compared to non-critically ill patients and healthy controls. ADAM9 is proposed to interfere with virus uptake and or replication. These findings were validated in a separate cohort and in vitro suppression of ADAM9 was shown to reduce SARS-CoV-2 replication in human epithelial cells in culture [97]. The top five genes—ADAM9, RAB10, MCEMP1, MS4A4A, and GCLM—found in this study were all significantly upregulated in critically ill patients [97]. These findings suggest that unique gene signatures may influence SARS-CoV-2 pathobiology and, hence, breakthrough infections in some COVID-19 patients. The risk of severe COVID-19 is often found among those individuals with chronic underlying medical conditions such obesity, diabetes mellitus, hypertension, cardiovascular disease, chronic kidney disease and liver disease. In a study Shin et al., that examined metabolic disorders and COVID-19 disease, they observed that in patients with hyperinsulinemia there was an increase in the expression of GRP78, which is a binding partner of the SARS-CoV-2 spike protein and ACE2 in adipocytes [98]. This increase expression of GRP78 could result in increased virus replication in adipocytes leading to increased viral loads, increased disease burden, and a higher rate of breakthrough infections. 

### 6.6. Other Factors Implicated in Breakthrough Infections

Gender could be predisposing factor to breakthrough infections. It is well known that females mount a more robust humoral and cellular immune responses than males but their susceptibility to COVID-19 breakthrough can vary depending their underlying comorbidities. In a study performed by Sun et al. that examined the association between immune dysfunction and COVID-19 breakthrough infection after COVID vaccinations in the US from the National COVID Cohort Collaborative, they found that gender was a factor. From a total of 664,722 patients with a median age of 51 years, they observed that individuals with a breakthrough infection after full vaccination were more likely to be older and women [99]. In addition, more breakthrough infections in this cohort were observed in individuals with HIV, solid organ transplants, and those with rheumatoid arthritis [99]. However, a smaller study performed by Basso et al. that examined breakthrough infections among health care workers was significantly associated with mild disease and men with diabetes that were overweight [100]. Even more those health care workers with higher levels of exposure to COVID patients were at higher risk of breakthrough infections [100]. 

Influenza vaccination has been implicated as being beneficially associated with reduced disease severity and mortality in COVID patients [101,102,103,104]. However, the direct effect of prior influenza vaccination in reducing the risk of COVID-19 breakthrough infections is unclear and requires further investigation. 

## 7. Strategies for Reducing Breakthrough Infections

Vaccinations/boosters and COVID-19 mitigation strategies, such as masking, ventilation, and social distancing, are among the strategies to curtail the incidence of breakthrough infections in the US. A measured approach to ending mask and vaccine mandates must be created. Innovative strategies must be developed to improve vaccine confidence by establishing public–private partnerships, such as the NIH Community Engagement Alliance against COVID-19 (CEAL), state and local health departments that include programs targeting vaccine hesitancy and resistance based on fear, misinformation, distrust of the government and medical establishment, conspiracy theories, antivaxxer propaganda, and vaccine politicization. 

The management of underlying comorbidities may support immune function and reduce the likelihood of having breakthrough infections. Individuals with weakened immune systems induced by underlying medical conditions or treatments for HIV, organ transplants, and chemotherapy for cancer and other diseases are encouraged to get a third shot or get boosted after consulting their doctor. This may mean, in some cases, a fourth shot or another booster. On a larger scale, we may reduce breakthrough infections by monitoring the emergence of new variants. To ensure that these breakthroughs are not endemic, coronavirus surveillance via whole genome sequencing programs must be a part of the toolbox of strategies to detect novel variants prior to the development of outbreaks that may contribute to increased breakthrough infections. Furthermore, it is essential that we monitor the potential emergence of novel variants via zoonotic intermediate hosts, such as the white-tailed deer, minks, and other domestic and wild animal species shown to be infected with SARS-CoV-2 variants [105,106,107]. 

We are also aware that the emergence of future variants is likely to occur in regions of the US and globally that remain largely unvaccinated and medically underserved. Many of these poor resource communities do not have access to COVID-19 vaccines or the resources to purchase vaccine contracts or have the health care infrastructure to support large-scale administration of vaccine especially for those living in rural communities. Vaccine hesitancy and resistance is still widespread in the US and around the world, which significantly impacts vaccination rates globally. In addition, we will have to consider new guidelines for future vaccines and their administration. Virus-specific vaccines and viral combinatorial vaccines for COVID-19 are currently being developed. Additional studies should be performed to extend the duration of immune protection after COVID-19 vaccinations that should involve the inclusion of novel FDA-approved adjuvants, proteins/antigens that would potentiate the immune response, new routes of administration (intranasal vs intramuscular) to achieve enhanced mucosal immunity, and vaccines that are more effective in older adults that are most vulnerable to most severe complication of COVID-19. These strategies could reduce the number of breakthrough infections over time. The infusion of undocumented immigrant populations has played a significant role in the dissemination of COVID-19 in underserved communities around the world. Efforts to vaccinate these populations have been difficult because of failed immigration policies. It is essential to develop a large-scale vaccination strategy to vaccinate immigrant populations regardless of their status. Immigrant communities should be provided with access to COVID-19 vaccines, vaccine information that is culturally competent, and COVID-19 related health services as part of their pathway to citizenship. Global COVID-19 vaccination programs targeting immigrant populations could impact the development of variants in these communities and reduce the burden of breakthrough infections.

## 8. Conclusions and Discussion

Although the Delta and Omicron variants have been shown to escape immune protection from the current vaccines [108,109], the mRNA-based COVID-19 vaccine boosters induce neutralizing immunity against the SARS-CoV-2 Omicron variant [110]. These findings support the importance of booster vaccinations. However, counties across the US with longstanding low COVID-19 vaccination rates, including counties in the South and Midwestern states, as well as urban communities with dense populations of Black and Hispanic/Latinx populations, continue to be impacted heavily by the Omicron variant. These infections will lead to an increase in breakthrough infections over time. The pandemic is now two years old, and the Omicron variant is of great concern and the predominant variant in the US and most countries around the world. The Omicron variant has been shown to be more transmissible than Delta but produces less severe disease, in general, but the severity of disease may vary among patients. 

COVID-19 variants will continue to develop. As the viruses infect animals and then transmitted to humans, this will add to the evolution of additional variants that could lead to other outbreaks and future pandemics. If the SARS-CoV-2 virus and resulting COVID-19 disease become endemic on a global scale, then we must endure the annual burden of disease and mortality as observed annually with influenza. In the US, some populations have developed significant immunity to this virus, yet this could be a source of selective pressure contributing to more variants. Even more, the virus being transmitted at high rates globally also will contribute to the development of new variants. 

Vaccinating the global population is essential for preventing and monitoring of breakthrough infections. Overall, new infections, hospitalizations, and deaths due to the rise of the Omicron variant have been overwhelming for states and counties throughout the US, specifically areas in the South. Breakthrough infections in the US currently are trending higher, immune protection due to the current vaccines offer some protection against the highly contagious Omicron variant, but vaccine hesitancy and resistance have resulted in stagnant vaccination rates. Inclusion of children ages 5–11 by the FDA EUA [111] and recent consideration of children ages 6 months to 4 years will require an aggressive vaccination campaign.

## Figures and Tables

**Figure 1 vaccines-10-00755-f001:**
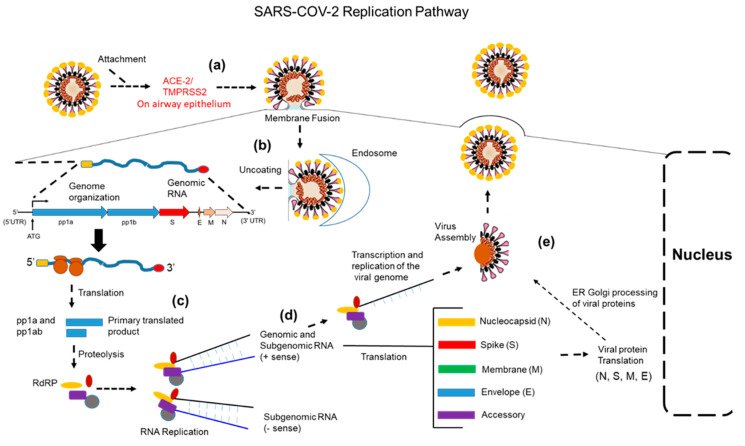
SARS-CoV-2 replication in human airway epithelium: (**a**) initiation of infection is accomplished via binding of the viral spike glycoprotein to ACE-2/TMPRSS2 (serine protease); (**b**) membrane fusion, endosomal uptake and release of the viral genomic RNA; (**c**) translation of the pp1a/pp1b loci results in a polyprotein that is proteolytically cleaved to produce several proteins that assemble into the RdRP/replication complex; (**d**) RNA replication proceeds producing both genomic and sub-genomic RNAs that are positive sense for viral genome packaging and for translation of viral structural proteins as well sub-genomic RNAs that serve as a template for making more positive genomic RNAs; (**e**) translations and ER/Golgi processing of viral structural proteins and replication of full-length viral genomes leads to virion assembly and egress.

**Figure 2 vaccines-10-00755-f002:**
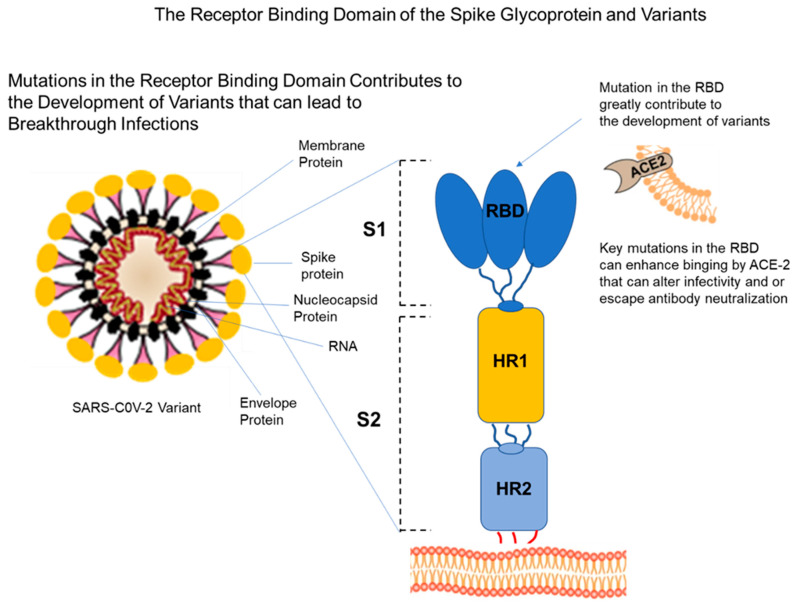
Mutations in the receptor-binding domain (RBD) of human coronaviruses contribute to the development of variants that may lead to breakthrough infections. These mutations in the RBD domain of the S1 subunit of the spike glycoprotein must bind to the ACE2 protein to initiate infection that can lead to viral transmission. The SARS-CoV-2 variant is shown and as well as the components of the virion structure that include the membrane protein, spike protein, nucleocapsid protein, the envelop protein, and RNA genome. The receptor-binding domain (RBD) of the S1 subunit of the spike protein is shown, which binds the ACE-2 protein to facilitate viral entry. The two heptad repeat (HR) regions of the S2 subunit HR1 and the HR2 are also shown.

**Figure 3 vaccines-10-00755-f003:**
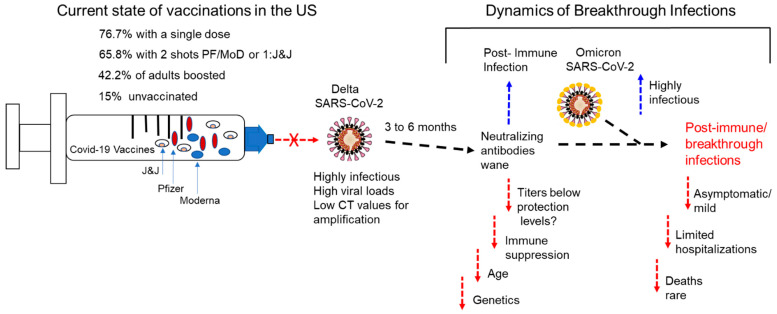
US vaccination rates, immune surveillance, and the dynamics of COVID-19 breakthrough infections. The current state of vaccination in the US including the percentage of single doses given, completed vaccinations, adults boosted, and the current unvaccinated population. The approved vaccines administered in the US (J&J Pfizer and Moderna) are shown. The delta variant is shown to have increased infectivity and a waning of immune protection after 3–6 months. The dynamics of breakthrough infections are shown to include conditions that are associated with breakthrough infections such as the loss of immune protection over time, immune suppression, age, and genetics. The potential impact of the omicron variant is shown, and it could contribute to breakthrough infection. Although highly infectious, it has resulted in mild disease, limited hospitalizations, and reduced mortality.

**Figure 4 vaccines-10-00755-f004:**
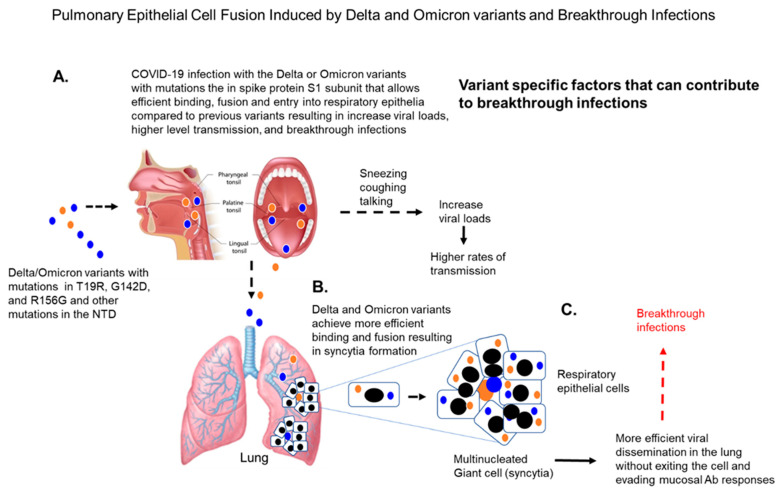
COVID-19 infections: both the Delta and Omicron variants have mutations in the N-terminal domain that can contribute to their transmission and dissemination resulting high viral loads (**A**). The Delta and Omicron variants under certain conditions can cause cell-to-cell fusion of respiratory epithelial cells resulting in syncytia formation (**B**). These syncytia or multinucleated giant cells can release infection virus to infect neighboring cells without exiting the cell, thereby evading mucosal antibody responses that could contribute to breakthrough infections (**C**).

**Figure 5 vaccines-10-00755-f005:**
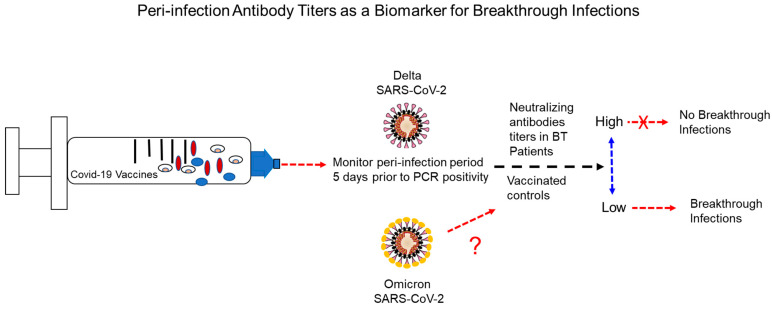
**Peri-infection antibody titers as a potential biomarker for breakthrough infections.** After the administration of FDA-approved vaccines in the US to health care workers, neutralizing antibody titers examined 5 days prior to PCR positivity (peri-infection period) compared in individuals with breakthrough infection (BT) and vaccinated controls. Individuals with high-neutralizing antibody titers during the peri-infection period were less likely to have a breakthrough infection. The full impact of the Omicron variant on breakthrough infections is unknown.

## Data Availability

The study did not report any laboratory-based data.

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
