# Peer review of "Breakthrough COVID-19 Infections in the US: Implications for Prolonging the Pandemic"

_vaccines, 2022, doi:10.3390/vaccines10050755_

Round 1

Reviewer 1 Report

Authors of this study evaluated the data of breakthrough Covid-19 infections in US, and discussed different factors which contribute to the occurrence of the breakthrough infections and its potential effect on duration of the pandemic. This topic is timely important. The study design is proper, and the structure of this review is logic. However, I have following concerns which should be addressed first before it can be published.

Major comments:

  1. In general, the factors causing breakthrough infections can be related to either pathogen or host or both. The first four chapters were focused on viruses and their variants e.g. Delta and Omicron variants, and the 5th and 6th chapters described the different host factors including preexisting individual conditions e.g. age, genetics and different underlying comorbidities. Although it is mentioned, there was no detailed discussion about what and how genetic factors could contribute to the breakthrough infections. It is important to add a (short) paragraph about this issue.
  2. The review is focused on data of breakthrough infections in US, especially when the Delta and Omicron emerged. Since Omicron appeared in last November, data collected in US may be limited. If such data from other countries are available, it would be important to include a summary.        

Minor comments include:

L44, As of March 13, should be written "As of March 13, 2022".

L218, The term "virus immune escape" should be changed to "virus escape". 

Author Response

April 22, 2022

Editor and Chief
Journal Vaccines

Manuscript ID vaccines-1659140

Dear Editor,

My responses to reviewers comments regarding manuscript Manuscript ID vaccines-1659140 entitled “Breakthrough COVID-19 Infections in the US: Implications for Prolonging the Pandemic are enclosed.

Thank you for giving me the opportunity to resubmit this manuscript to your journal for publication.

Kind regards,

Reviewer 1

Major comments:1. In general, the factors causing breakthrough infections can be related to either pathogen or host or both. The first four chapters were focused on viruses and their variants e.g. Delta and Omicron variants, and the 5th and 6th chapters described the different host factors including preexisting individual conditions e.g. age, genetics and different underlying comorbidities. Although it is mentioned, there was no detailed discussion about what and how genetic factors could contribute to the breakthrough infections. It is important to add a (short) paragraph about this issue.

Authors’ response:  We agree with the reviewer and have added information to support genetic factors that could contribute to the breakthrough infections. The information appears in blue text in a new section entitled “Genetics and Breakthrough Infections.” 

  1. The review is focused on data of breakthrough infections in US, especially when the Delta and Omicron emerged. Since Omicron appeared in last November, data collected in US may be limited. If such data from other countries are available, it would be important to include a summary.    

Authors’ response: We agree with the reviewer, but I was not able to find any substantive data on breakthrough infection with the Omicron variant globally that has been published in PubMed, other than a paper documenting Omicron specific breakthrough infections in a small group of German visitors to South Africa. This correspondence report only involved 7 individuals.

Reference: Kuhlmann C, Mayer CK, Claassen M, Maponga T, Burgers WA, Keeton R, Riou C, Sutherland AD, Suliman T, Shaw ML, Preiser W. Breakthrough infections with SARS-CoV-2 omicron despite mRNA vaccine booster dose. Lancet. 2022 Feb 12;399(10325):625-626.

Minor comments include:

L44, As of March 13, should be written "As of March 13, 2022".

The statement has been changed and appears in blue text in the revised manuscript.

L218, The term "virus immune escape" should be changed to "virus escape". 

The statement has been changed and appears in blue text in the revised manuscript.

Reviewer 2 Report

In their review, Alcendor et al. discuss aspects of SARS-CoV-2 breakthrough infection with respect to prolonging the pandemic situation in the US. The topic of the review is very timely as especially in developed countries the majority of the population has already been immunized either by vaccines or by natural infections.

Major concerns:

Early variants of concerns before omicron including the delta variant were demonstrated to be efficiently neutralized by sera from vaccinated individuals – even if the neutralization of delta is less efficient. In contrast, recent data show a weak neutralization capacity of omicron with sera of vaccinated individuals. Booster vaccination enhances the neutralizing capacity. The best neutralization was found in vaccinated people with a history of natural infection. Thus “classical” breakthrough infections related to waning immunity, not fully immunization, age, or other factors are rather relevant for earlier VoCs but not for omicron. There is growing evidence from European countries that vaccination is not really protective against omicron, which is obvious from high incidences despite high vaccination rates in several countries. The concept of herd immunity by vaccination (with current vaccines) seems to be fallen by omicron. In this point of view, I would suggest to avoid the term “breakthrough” infections in case of omicron or at least discussing it in more detail based on the newest data available.    

In contrast to neutralizing capacity of antibodies, cellular response induced by natural infections or vaccinations seems to be not affected by mutations - see studies of the Sette lab and others. However, cellular responses are not playing a substantial role in protection from infection, but due to its potential involvement in the reduction of severe outcomes of COVID-19 it should also be discussed.

Minor points:

SARS-CoV-2 represents an RNA virus with proofreading capacity. Thus, mutations happen from time to time but they occur not so frequent as in the case of other RNA viruses without repair mechanisms. This should be included in chapter 3.

Repeating information should be skipped throughout the text.

Figure legends should clearly explain the respective figures. Some of the figures are too busy.

Author Response

April 22, 2022

Editor and Chief
Journal Vaccines

Manuscript ID vaccines-1659140

Dear Editor,

My responses to reviewers comments regarding manuscript Manuscript ID vaccines-1659140 entitled “Breakthrough COVID-19 Infections in the US: Implications for Prolonging the Pandemic are enclosed.

Thank you for giving me the opportunity to resubmit this manuscript to your journal for publication.

Kind regards,

Reviewer #2

Major concerns:

Early variants of concerns before omicron including the delta variant were demonstrated to be efficiently neutralized by sera from vaccinated individuals – even if the neutralization of delta is less efficient. In contrast, recent data show a weak neutralization capacity of omicron with sera of vaccinated individuals. Booster vaccination enhances the neutralizing capacity. The best neutralization was found in vaccinated people with a history of natural infection. Thus “classical” breakthrough infections related to waning immunity, not fully immunization, age, or other factors are rather relevant for earlier VoCs but not for omicron. There is growing evidence from European countries that vaccination is not really protective against omicron, which is obvious from high incidences despite high vaccination rates in several countries. The concept of herd immunity by vaccination (with current vaccines) seems to be fallen by omicron. In this point of view, I would suggest to avoid the term “breakthrough” infections in case of omicron or at least discussing it in more detail based on the newest data available.

Authors’ response: We agree with the reviewer and have added information in blue text in the Omicron variant section of the revised manuscript.

In contrast to neutralizing capacity of antibodies, cellular response induced by natural infections or vaccinations seems to be not affected by mutations - see studies of the Sette lab and others. However, cellular responses are not playing a substantial role in protection from infection, but due to its potential involvement in the reduction of severe outcomes of COVID-19 it should also be discussed.

Authors’ response. We agree with the reviewer, and have added information to the Aging and immune responses to vaccines section of the revised manuscript in blue text.

Minor points:

SARS-CoV-2 represents an RNA virus with proofreading capacity. Thus, mutations happen from time to time but they occur not so frequent as in the case of other RNA viruses without repair mechanisms. This should be included in chapter 3. 

Authors’ response: We agree with the reviewer, and have added this information to the revised manuscript in section #3.  Reference: Ma Z, Pourfarjam Y, Kim IK. Reconstitution and functional characterization of SARS-CoV-2 proofreading complex. Protein Expr Purif. 2021 Sep;185:105894.

Repeating information should be skipped throughout the text.

Authors’ response: We some what agree with the reviewer however some level of repetition is normal in this context.  

Figure legends should clearly explain the respective figures. Some of the figures are too busy. 

Authors’ response:  We somewhat agree with the reviewer and have added information to the figure to achieve a fuller explanation of the figures. However, the busy figure is designed to provide a better understanding of mechanisms we are proposing that will be beneficial to a broader reading audience. In addition, we have expanded the figure legends to include more information.

Reviewer 3 Report

The manuscript gives a general outlook off what the SARS CoV2 pandemic has been up to the moment in which the article was written. This is one of many reviews stating more on less the same issues, asking the same questions without providing real guidance to the readers. The authors did not discuss the importance of several factors which may be relevant, influenza vaccination, age and gender predisposition of the disease or serious secondary effects of vaccines. The broader scope of the review gives the reader empty-handed without answering the question raised in the title. A shorter descriptive perspective may be reasonable, but includes comments in future virus outbreaks. Three questions may be addressed: 1)  the generation of new variants could be possible in areas of the US with low vaccination/low healthy conditions? 2) What will be reasonable guidelines for vaccines, change actual vaccine intramuscular administration to intranasal administration? In this case will protein-based vaccines increase adherence of repetitive vaccination?, 3) A high number of non legal immigrants can hamper any epidemiological guidelines, what do the authors consider that could be relevant in this population. In summary, novel arguments have to be introduced to the manuscript.

Author Response

April 22, 2022

Editor and Chief
Journal Vaccines

Manuscript ID vaccines-1659140

Dear Editor,

My responses to reviewers comments regarding manuscript Manuscript ID vaccines-1659140 entitled “Breakthrough COVID-19 Infections in the US: Implications for Prolonging the Pandemic are enclosed.

Thank you for giving me the opportunity to resubmit this manuscript to your journal for publication.

Kind regards,

Reviewer #3

The manuscript gives a general outlook off what the SARS CoV2 pandemic has been up to the moment in which the article was written. This is one of many reviews stating more on less the same issues, asking the same questions without providing real guidance to the readers. The authors did not discuss the importance of several factors which may be relevant, influenza vaccination, age and gender predisposition of the disease or serious secondary effects of vaccines. The broader scope of the review gives the reader empty-handed without answering the question raised in the title. A shorter descriptive perspective may be reasonable, but includes comments in future virus outbreaks. Three questions may be addressed: 1)  the generation of new variants could be possible in areas of the US with low vaccination/low healthy conditions?

Authors’ response We agree with the reviewer that the generation of new variants could be possible in areas of the US with low vaccination/low healthy conditions. We have included information in the revised manuscript in blue text to address the reviewer’s comments in the section “Strategies for reducing breakthrough infections.”

2) What will be reasonable guidelines for vaccines, change actual vaccine intramuscular administration to intranasal administration? In this case will protein-based vaccines increase adherence of repetitive vaccination.

Authors’ response: We have included information in the revised manuscript in blue text to address the reviewer’s comments in the section “Strategies for reducing breakthrough infections.”

3) A high number of non-legal immigrants can hamper any epidemiological guidelines, what do the authors consider that could be relevant in this population. In summary, novel arguments have to be introduced to the manuscript.

Authors’ response: We have included information in the revised manuscript in blue text to address the reviewer’s comments in the section “Strategies for reducing breakthrough infections.”

Reviewer 4 Report

The manuscript tried to review breakthrough COVID-19 infections with Delta and Omicron variants of SARS-CoV-2 in the US. The manuscript needs vast revision.

Authors emphasized the need of expedite vaccination. For example, “Although the actual effectiveness of COVID-19 vaccines against the Omicron variant is unclear, current evidence indicates that vaccines authorized in the United States provide substantial level of protection against this variant.” (Lines 442-443). But, Delta and Omicron variants were known to escape protection from the current vaccination. Authors should show references to support their statement. In addition, the content of the manuscript is sometimes duplicated and not organized.

Figure numbering and legends are out of order. Correct them.

References: Many references are duplicated. Check the references carefully before submission.

Line 558. Access time is too old and should be revised.

Line 700. Ref#57 (Shah et al., 2022) is the same as Ref#35 (Shah et al., 2022).

Line 703. Ref#58 (Bushman et al., 2021) is the same as Ref#13 (Bushman et al., 2021).

Line 717. Ref#61 (Chen et al., 2022) is basically the same as Ref#59, isn’t it?

Line 737. Ref#67 (CNN 2021/04/14) is the same as Ref#14 (CNN 2021/04/14).

Line 740. Ref#68 (Juthani et al., 2021) is the same as Ref#15 (Juthani et al., 2021).

Line 773. Ref#77 (Chen et al., 2021) is the same as Ref#75 (Chen et al., 2021)!

Line 800. Access time is too old and should be revised.

Other minor comments:

Line 44. “March 13” should be “March 13, 2022”.

Line 124. “isaccomplished” should be “is accomplished”.

Lines 186, 187. Explain the meaning of asterisks.

Line 199. “Omicron” reads “Delta”. 

Author Response

April 22, 2022

Editor and Chief
Journal Vaccines

Manuscript ID vaccines-1659140

Dear Editor,

My responses to reviewers comments regarding manuscript Manuscript ID vaccines-1659140 entitled “Breakthrough COVID-19 Infections in the US: Implications for Prolonging the Pandemic are enclosed.

Thank you for giving me the opportunity to resubmit this manuscript to your journal for publication.

Kind regards,

Reviewer #4 Comments

The manuscript tried to review breakthrough COVID-19 infections with Delta and Omicron variants of SARS-CoV-2 in the US. The manuscript needs vast revision.

Authors emphasized the need of expedite vaccination. For example, “Although the actual effectiveness of COVID-19 vaccines against the Omicron variant is unclear, current evidence indicates that vaccines authorized in the United States provide substantial level of protection against this variant.” (Lines 442-443). We realize that statement represented an earlier finding.  The statement has been removed and updated to reflect the immune escape capacity of the Delta and Omicron variants 

But, Delta and Omicron variants were known to escape protection from the current vaccination. Authors should show references to support their statement. In addition, the content of the manuscript is sometimes duplicated and not organized.

Authors’ response.  Some level of duplications is warranted for the general reading audience but has been held to a minimum.

We realize that the statement represented an earlier finding.  The statement has been removed and updated to reflect the immune escape capacity of the Delta and Omicron variants. References have been included to support the revised statement.

Although the Delta and Omicron variants have been shown to escape immune protection from the current vaccines [91], the mRNA-based COVID-19 vaccine boosters induce neutralizing immunity against the SARS-CoV-2 Omicron variant [91, 92].  These findings support the importance booster vaccinations.

Reference #91. Dejnirattisai W, Huo J, Zhou D, Zahradník J, Supasa P, Liu C, Duyvesteyn HME, Ginn HM, Mentzer AJ, Tuekprakhon A, Nutalai R, Wang B, Dijokaite A, Khan S, Avinoam O, Bahar M, Skelly D, Adele S, Johnson SA, Amini A, Ritter TG, Mason C, Dold C, Pan D, Assadi S, Bellass A, Omo-Dare N, Koeckerling D, Flaxman A, Jenkin D, Aley PK, Voysey M, Costa Clemens SA, Naveca FG, Nascimento V, Nascimento F, Fernandes da Costa C, Resende PC, Pauvolid-Correa A, Siqueira MM, Baillie V, Serafin N, Kwatra G, Da Silva K, Madhi SA, Nunes MC, Malik T, Openshaw PJM, Baillie JK, Semple MG, Townsend AR, Huang KA, Tan TK, Carroll MW, Klenerman P, Barnes E, Dunachie SJ, Constantinides B, Webster H, Crook D, Pollard AJ, Lambe T; OPTIC Consortium; ISARIC4C Consortium, Paterson NG, Williams MA, Hall DR, Fry EE, Mongkolsapaya J, Ren J, Schreiber G, Stuart DI, Screaton GR. SARS-CoV-2 Omicron-B.1.1.529 leads to widespread escape from neutralizing antibody responses. Cell. 2022 Feb 3;185(3):467-484.e15.

Reference #92 Hoffmann, M. et al. SARS-CoV-2 variant B.1.617 is resistant to Bamlanivimab and evades antibodies induced by infection and vaccination. Cell Rep. https://doi.org/10.1016/ j.celrep.2021.109415 (2021).

Reference #93.  Schmidt AG, Iafrate AJ, Naranbhai V, Balazs AB. mRNA-based COVID-19 vaccine boosters induce neutralizing immunity against SARS-CoV-2 Omicron variant. Cell. 2022 Feb 3;185(3):457-466.e4. doi: 10.1016/j.cell.2021.12.033. Epub 2022 Jan 6. PMID: 34995482; PMCID: PMC8733787.

Figure numbering and legends are out of order. Correct them. 

Authors response. The disorder observed in the figure legends is part of the journal processing of the manuscript.

References: Many references are duplicated. Check the references carefully before submission.

Line 558. Access time is too old and should be revised.

Author’s response. Accessed time has been updated and revised

Line 700. Ref#57 (Shah et al., 2022) is the same as Ref#35 (Shah et al., 2022). Authors’ response. Reference #35 has been replaced.

Line 703. Ref#58 (Bushman et al., 2021) is the same as Ref#13 (Bushman et al., 2021).

Authors’ response. Reference #13 has been replaced.

Line 717. Ref#61 (Chen et al., 2022) is basically the same as Ref#59, isn’t it? Authors’ response. Reference #61 has been replaced.

Line 737. Ref#67 (CNN 2021/04/14) is the same as Ref#14 (CNN 2021/04/14).

Authors’ response Reference #14 has been replaced.

Line 740. Ref#68 (Juthani et al., 2021) is the same as Ref#15 (Juthani et al., 2021). Authors’ response. Reference #15 has been replaced.

Line 773. Ref#77 (Chen et al., 2021) is the same as Ref#75 (Chen et al., 2021)! Authors’ response. Reference #75 has been replaced.

Line 800. Access time is too old and should be revised.

Author’s response. Accessed time has been updated and revised

Other minor comments:

Line 44. “March 13” should be “March 13, 2022”.

The change has been made and appears in blue text

Line 124. “isaccomplished” should be “is accomplished”.

The change has been made and appears in blue text

Lines 186, 187. Explain the meaning of asterisks.

The asterisk means that these mutations are exclusive present in the Delta Plus variant.  This information has been included in the revised manuscript in blue text.

Line 199. “Omicron” reads “Delta”. 

The change has been made and appears in blue text

Round 2

Reviewer 3 Report

The authors made some changes in the manuscript, but there are important issues non responded to by the authors. 

Author Response

April 27, 2022

Editor and Chief
Journal Vaccines

Manuscript ID vaccines-1659140

Dear Editor,

My responses to reviewers comments regarding manuscript Manuscript ID vaccines-1659140 entitled “Breakthrough COVID-19 Infections in the US: Implications for Prolonging the Pandemic are enclosed.

Thank you for giving me the opportunity to resubmit this manuscript to your journal for publication.

Kind regards,

Donald J. Alcendor, Ph.D.

Associate Professor

Meharry Medical College

Center for AIDS Health Disparities Research

& Department of Microbiology and Immunology

& Obstetrics and Gynecology

Hubbard Hospital 5th Floor Rm. 5025

1005 Dr. D.B. Todd Jr. Blvd.

Nashville, TN 37208

Phone: 615-327-6449

Fax: 615-327-6929

Email: dalcendor@mmc.edu

Associate Professor Adjunct

Department of Pathology, Microbiology and Immunology

Vanderbilt University Medical Center

Reviewer #3

The manuscript gives a general outlook off what the SARS CoV2 pandemic has been up to the moment in which the article was written. This is one of many reviews stating more on less the same issues, asking the same questions without providing real guidance to the readers. The authors did not discuss the importance of several factors which may be relevant, influenza vaccination, age and gender predisposition of the disease or serious secondary effects of vaccines. The broader scope of the review gives the reader empty-handed without answering the question raised in the title. A shorter descriptive perspective may be reasonable, but includes comments in future virus outbreaks. Three questions may

The authors did not discuss the importance of several factors which may be relevant, influenza vaccination, age and gender predisposition of the disease or serious secondary effects of vaccines.

Authors’ response: We agree with the reviewer and have provided information in response to the comments in red text in the revised manuscript. We have already responded age and the risk of breakthrough infections in the Aging and immune responses to vaccines section of the manuscript. The section is high-lighted in red text. References to the added information can be found in the reference section in red text. We have provided information on the relevance of Influenza vaccinations and COVID-19 disease severity that is not directly linked to breakthrough infections.  We have seen no substantive studies in the PubMed databased that directly links side effects of the COVID-19 vaccine directly to the risk of breakthrough infections. 

be addressed: 1)  the generation of new variants could be possible in areas of the US with low vaccination/low healthy conditions?

Authors’ response We agree with the reviewer that the generation of new variants could be possible in areas of the US with low vaccination/low healthy conditions. We have included information in the revised manuscript in blue text to address the reviewer’s comments in the section “Strategies for reducing breakthrough infections.”

2) What will be reasonable guidelines for vaccines, change actual vaccine intramuscular administration to intranasal administration? In this case will protein-based vaccines increase adherence of repetitive vaccination.

Authors’ response: We have included information in the revised manuscript in blue text to address the reviewer’s comments in the section “Strategies for reducing breakthrough infections.”

3) A high number of non-legal immigrants can hamper any epidemiological guidelines, what do the authors consider that could be relevant in this population. In summary, novel arguments have to be introduced to the manuscript.

Authors’ response: We have included information in the revised manuscript in blue text to address the reviewer’s comments in the section “Strategies for reducing breakthrough infections.”
